# Rethinking Low-Confidence Pseudo Labels: Influence-Aware Semi-Supervised Fine-Tuning for Hyperspectral Change Detection

Keyun Zhao [1]   Guangchang Li [2]   Yunpeng Bai [2]   Jiang Shao [1]   Ying Li [1]

## Abstract

Hyperspectral image change detection (HSI-CD) suffers from severe annotation scarcity and complex change patterns, which fundamentally limit the effectiveness of directly fine-tuning pre-trained foundation models. Although semi-supervised learning provides a promising direction, existing approaches mainly rely on confidence-based pseudo-label selection, leading to limited data diversity or severe error propagation. In this paper, we propose Influence-Aware Semi-supervised Fine-tuning (IA-SFT), a novel framework that evaluates the influence of pseudo-labels on model decision behavior to identify truly valuable supervision signals. Instead of confidence-based selection, IA-SFT evaluates each low-confidence pseudo-label by measuring its impact on labeled data, enabling reliable filtering of high-value pseudo-labels with minimal noise. To further adapt foundation models to HSI-CD, we design an Adaptive Fusion Change Decoder (AFCD) that jointly models global semantic consistency and local change details. Extensive experiments on three benchmark datasets demonstrate that IA-SFT consistently improves pseudo-label quality and detection performance, achieving superior accuracy compared to state-of-the-art methods. Additional analyses validate the transferability of IA-SFT when integrated into different frameworks in a plug-and-play manner. Code will be released.

[1]School of Computer Science, Northwestern Polytechnical University, Xi'an 710029, China [2]School of Software, Northwestern Polytechnical University, Xi'an 710029, China. Correspondence to: Yunpeng Bai <cloudbai@nwpu.edu.cn>, Ying Li <lybyp@nwpu.edu.cn>.

*Proceedings of the 43rd International Conference on Machine Learning*, Seoul, South Korea. PMLR 306, 2026. Copyright 2026 by the author(s).

## 1. Introduction

Hyperspectral images (HSIs), with rich spectral information, are highly advantageous for monitoring land-cover categories and dynamic surface processes. Among the various interpretation tasks based on HSIs, HSI change detection (HSI-CD) aims to identify the differences between bi-temporal HSIs, thereby enabling an accurate characterization of dynamic changes on the Earth's surface (Lv et al., 2025). With the rapid development of imaging and sensing technologies, HSI-CD has been widely applied in fields such as land-cover updating (Guo et al., 2021), urban change analysis (Huang et al., 2019), and military reconnaissance (Ertürk et al., 2017).

In general, previous deep learning–based HSI-CD methods can be categorized from multiple perspectives, including technical paradigms, network architectures, and learning strategies. From the viewpoint of technical paradigms, current methods mainly consist of convolutional neural network (CNN)–based methods (Lu et al., 2024), Transformer-based methods (Wang et al., 2022b), and graph neural network (GNN)–based methods (Xu et al., 2024), etc. The differences of these methods lie in their different emphasizes on local and global correlations. Depending on architectures, previous methods can be divided into single-stream (Song et al., 2022), dual-stream (Luo et al., 2023), and three-stream architectures (Zhao et al., 2023). The architectural differences are directly manifested in variations of model complexity as well as representational ability. Moreover, according to different learning strategies, previous methods can be classified into fully supervised (Ding et al., 2024), semi-supervised (Qu et al., 2024), and unsupervised methods (Wang et al., 2025b). The selection of learning strategies is mainly determined by the availability of labeled samples.

In recent years, with the rapid development of foundation models in computer vision, their powerful representation learning ability provides a new paradigm for remote sensing change detection (Li et al., 2025). By adopting the paradigm of "pre-training + task-specific fine-tuning," the model can learn common features and take full advantage of domain knowledge. However, due to the intrinsic characteristics of HSI, this paradigm still faces two critical challenges:

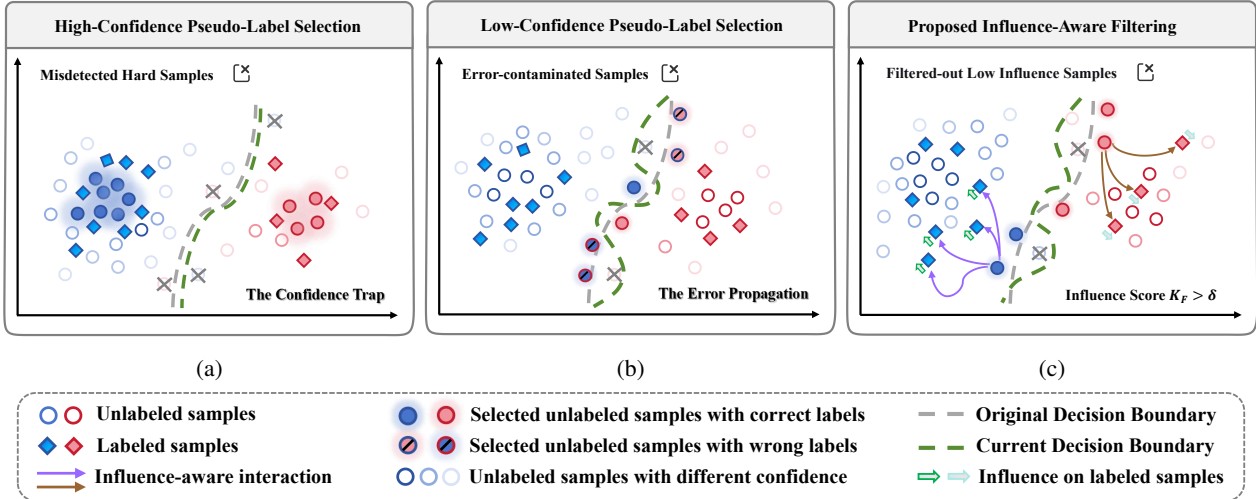

*Figure 1.* Illustration of different Confidence-based pseudo-label selection strategy. (a) High-confidence selection. (b) Low-confidence selection. (c) Proposed influence-aware filtering.

• Due to the high annotation cost of HSI-CD tasks, the scale of labeled samples is usually limited, under which direct fine-tuning is highly prone to overfitting. Meanwhile, introducing pseudo-labels for semi-supervised learning poses a dilemma. We illustrate this in Figure 1: high-confidence pseudo-labels may further aggravate overfitting, while low-confidence pseudo-labels tend to introduce noisy supervision, leading to error accumulation and propagation.

• HSI-CD are typically characterized by discontinuous change boundaries and sparsely distributed changes, which impose high requirements on the joint modeling of multi-scale semantic information and fine-grained change details. However, most existing foundation models are pre-trained for generic vision tasks, and their decoders are usually fixed in structure, lacking the adaptability required for hyperspectral change detection. Therefore, the application potential of foundation models are limited.

In this paper, to address the above issues, we investigate IAF-CDNet, which integrates an Influence-Aware Semi-supervised Fine-tuning (IA-SFT) strategy with an Adaptive Fusion Change Decoder (AFCD). Compared with previous methods, our contributions can be summarized as follows:

• We propose a pluggable pseudo-label selection strategy that goes beyond confidence-based selection. By evaluating the influence of pseudo-labels on the model's decision making, IA-SFT enables a more reliable selection of high-value unlabeled samples.

• We design an Adaptive Fusion Change Decoder that integrates global semantic context and local detail features through multi-level feature interactions, enabling more precise modeling of fine-grained change regions.

• We conduct comprehensive experiments across diverse models and benchmarks. The results demonstrate that IA-SFT identifies high-value pseudo-labels to boost change detection performance, exhibiting strong transferability and generalization capability.

## 2. Related Works

This section briefly reviews the topics related to this work, including semi-supervised HSI-CD Methods and pseudo-label selection strategies.

### 2.1. Semi-supervised HSI-CD Methods

Due to the high annotation cost of HSIs, semi-supervised learning has attracted increasing attention in HSI-CD. Meanwhile, owing to the constrained data scale and the commonly adopted dataset partition methods, HSI-CD methods typically follow a transductive semi-supervised setting, where unlabeled samples are drawn from the test set. Existing semi-supervised HSI-CD approaches mainly exploit unlabeled data through two representative paradigms. The first paradigm is consistency regularization. For example, Hyper-Match (Huang et al., 2024) and DCENet (Luo et al., 2024) apply perturbations of different intensities to unlabeled samples and optimize the model by jointly leveraging prediction consistency on unlabeled samples and supervised loss on the real samples. MSCSCNet (Qu et al., 2024) employs a mean teacher architecture to exploit unlabeled data. Other methods such as DAFormer (Wang et al., 2023) and CSR-Net (Liu & Sun, 2025) implement semi-supervised learning through self-supervised strategies, first training a feature extractor with a large number of unlabeled samples and then training a classification head using few labeled samples.

Despite encouraging progress, previous methods still suffer from notable limitations. Most approaches rely on confidence-based heuristics, consistency constraints, or uniform utilization of unlabeled samples, which makes them vulnerable to noisy pseudo-labels and limits their ability to exploit hard or ambiguous change samples effectively. Moreover, the potential influence of individual unlabeled samples on the learning process is rarely considered, leading to suboptimal utilization of low-confidence data. These limitations motivate the development of more principled strategies that can selectively leverage high-value unlabeled samples for robust semi-supervised change detection.

### 2.2. Pseudo-Label Selection Strategy

Pseudo-label–based semi-supervised learning has been extensively studied owing to its favorable transferability and empirical effectiveness. Early methods iteratively incorporate high-confidence pseudo-labeled samples into training, but their susceptibility to noisy predictions often results in error accumulation. To alleviate this issue, confidence-based selection strategies have been extensively explored, including FixMatch (Sohn et al., 2020), FlexMatch (Zhang et al., 2021), FreeMatch (Wang et al., 2022a), etc. While these approaches improve robustness compared to fixed-threshold strategies, they still fundamentally rely on prediction confidence as the primary criterion, making them vulnerable to over-confident yet erroneous pseudo-labels.

To overcome the limitations of static confidence thresholds, recent studies explored influence-based selection strategies, which can be categorized into three paradigms: Gradient-based influence (Koh & Liang, 2017; Liu et al., 2021; Pruthi et al., 2020), which evaluates the expected change in model parameters via gradient inner products; Data value influence (Ghorbani & Zou, 2019), which employs game-theoretic tools like Shapley values to assess a sample's marginal contribution to overall accuracy; and Representation influence (Zou & Zare, 2016), which measures how a sample alters spectral reconstruction. However, the reliability of these influence estimations is often heavily contingent upon the model's initial performance and exhibits significant instability in low-confidence regions, where volatile gradients may lead to suboptimal sample selection.

More recently, training-coupled strategies have been proposed to assess pseudo-label quality from an optimization perspective. Methods such as CPL (Zhang et al., 2024), CGMatch (Cheng et al., 2025), and SemiReward (Li et al., 2023) highlight an important insight: pseudo-label quality should not be judged solely by prediction confidence, but also by its influence on the learning process. In summary, existing pseudo-labeling strategies primarily focus on confidence, uncertainty, or learning dynamics as selection criteria. While each line of work improves robustness to noisy supervision to some extent, effectively identifying high-value pseudo-labeled samples—particularly those residing in ambiguous or low-confidence regions—remains a challenging problem. This limitation motivates the development of mechanisms that account for sample influence during training, which is the focus of our work.

## 3. Method

In this section, we present the proposed IAF-CDNet, a foundation-model-based HSI-CD framework that integrates an Influence-Aware Semi-supervised Fine-tuning (IA-SFT) strategy with an Adaptive Fusion Change Decoder (AFCD).

### 3.1. Overview

As illustrated in Figure 2, the pipeline of IAF-CDNet consists of five steps: 1) Perform supervised fine-tuning using labeled data $D_L$ to obtain an initial model $\theta_S$, which is then used to generate pseudo-labels for unlabeled data $D_U$; 2) Rank the pseudo-labels $P_U$ based on confidence, and select a portion of samples from both the high-confidence and low-confidence ranges, resulting in $D_H$ and $D_A$; 3) For each low-confidence sample $S_i \in D_A$, we conduct a sample-wise rapid fine-tuning. The tuned model $\theta_f^i$ is used to generate post-tuning predictions $P_O$ of labeled data $D_L$. Meanwhile, the initial model $\theta_S$ is used to generate the pre-tuning predictions $P_R$; 4) The influence of each low-confidence sample $S_i$ is evaluated by measuring the prediction shift $\triangle P$ between its pre-tuning and post-tuning predictions; 5) High-value pseudo-labels $D_V$ are selected based on the sample influence matrix $M_I$. In addition, high-confidence samples $D_H$ from Step 2 are used for class distribution calibration. Finally, the model undergoes the final semi-supervised fine-tuning using both high-value pseudo-labels $D_V$, distribution calibration samples $D_C \in D_H$ and labeled data $D_L$.

### 3.2. Influence-Aware Semi-supervised Fine-tuning

#### 3.2.1. MODEL AND DATA PREPARATION

In this work, we adopt the SpatialSigma encoder (Wang et al., 2025a) as the backbone, which is a vision transformer-based foundation model. Unlike the full-parameter fine-tuning strategy used in HyperSigma, we freeze the encoder and apply low-rank adaptation (LoRA) (Hu et al., 2022) for lightweight tuning, while further introducing a trainable decoder to improve the model's sensitivity to change information.

As illustrated in Step 1 of Fig. 2, the small-scale labeled data $D_L$ is first fed into the model for supervised fine-tuning, where the parameters are updated using the cross-entropy loss. After fine-tuning, all unlabeled data $D_U$ is inferred by model $\theta_S$ to generate pseudo-labels $P_U$.

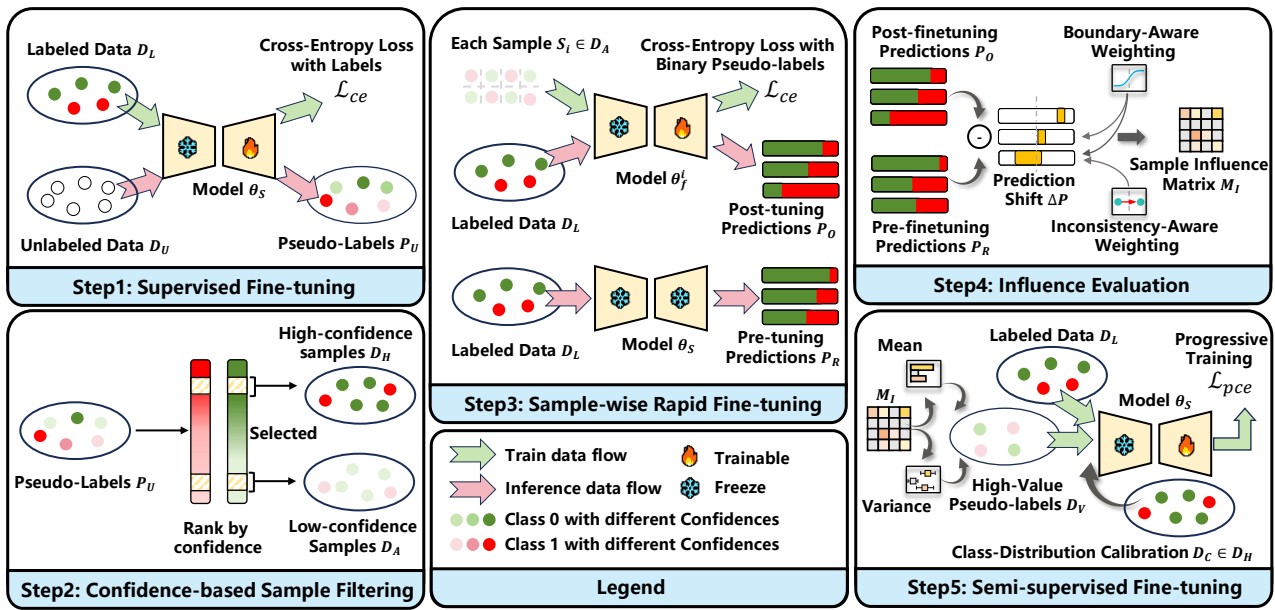

*Figure 2.* Pipeline of IAF-CDNet.

Unlike most existing methods that indiscriminately utilize all unlabeled samples for training, we first filter the unlabeled data according to prediction confidence and select subsets from both high- and low-confidence levels as candidate samples, resulting in $D_H$ and $D_A$. Samples with the highest and lowest confidence levels are excluded to mitigate excessive model overconfidence and to avoid the adverse effects caused by extremely unreliable pseudo-labels. In the subsequent process, $D_H$ is used for class distribution calibration, while $D_A$ serves as the primary focus of IA-SFT.

### 3.2.2. SAMPLE-WISE RAPID FINE-TUNING

Distinct from the gradient-based or representation-based paradigms discussed in Section 2.2, we define influence as the empirical shift in predictions on labeled data. We draw inspiration from the mirrored influence hypothesis (Ko et al., 2024) and perform sample-wise rapid fine-tuning via each sample $S_i \in D_A$. The tuned model $\theta_f^i$ and the initial model $\theta_S$ then infer on labeled data $D_L$ to generate post- and pre-tuning predictions $P_O$ and $P_R$. This procedure enables us to quantify how each pseudo-label influences the model's decision, thereby facilitating a more precise analysis of pseudo-labels.

### 3.2.3. INFLUENCE EVALUATION

To evaluate the influence of pseudo-labels, we analyze the prediction shift induced on the labeled data before and after fine-tuning. Specifically, we compute the prediction shift $\triangle P$ as the absolute value of the difference between the post-tuning predictions $P_O$ and the pre-tuning predictions $P_R$. However, the same magnitude of $\triangle P$ may carry

very different implications depending on where the shift occurs, particularly when the perturbation lies near the decision boundary. To address this, we introduce boundary-and inconsistency-aware weighting.

For boundary-aware weighting, the goal is to emphasize prediction shifts that occur near the decision boundary—i.e., the closer the pre-tuning prediction $P_R$ to the decision boundary, the more significant the influence of sample should be. To achieve this, we adopt a sigmoid-based non-linear weighting scheme, which can be formulated as:

$$W_B = \frac{1}{1 + \log(10 \times (z + Max(|P_R - 0.5|, b_h)))} \quad (1)$$

where $Max(\cdot)$ returns the greater value of the two operands; $b_h$ denotes a boundary margin within which the samples are treated as equally boundary-sensitive, and the boundary weight is intentionally saturated to its maximum value; $z$ is a small positive constant (set to 1e-6).

For inconsistency-aware weighting, the goal is to assign a bonus score to cases where the predicted class changes after fine-tuning. This process can be formulated as follows:

$$K_I = \begin{cases} 1, & (P_R - 0.5)(P_O - 0.5) < 0 \\ 0, & \text{otherwise} \end{cases} \quad (2)$$

where $K_I$ denotes the resulting inconsistency reward score. With the above components, we compute the final influence score for each pseudo sample as:

$$K_F = \triangle P \times (1 + W_B) + K_I \quad (3)$$

All influence scores between low-confidence samples $D_A$ and labeled samples $D_L$ are organized in a matrix form, yielding the sample influence matrix $M_I \in \mathbb{R}^{N_L \times N_A}$, where $N_L$ denotes the number of labeled samples and $N_A$ denotes the number of selected low-confidence samples.

### 3.2.4. SEMI-SUPERVISED FINE-TUNING

After influence evaluation, we proceed to select high-value pseudo-labels. Specifically, for each sample $S_i \in D_A$, we compute the mean and variance of its influence scores on all labeled samples, and we retain those whose mean and variance both fall within the top $\delta\%$. These labels are regarded high-value pseudo-labels $D_V$. The rationale is that a large mean influence indicates a strong and consistent impact on the decision boundary, while a large variance reflects diverse sensitivity across different labeled samples. These characteristics suggest a greater potential to enrich the data distribution and provide informative supervisory signals. Moreover, this strategy will also greatly improve the accuracy of low-confidence pseudo-labels.

However, the class distribution of $D_V$ often deviates considerably from the real class distribution. To address this, we apply the class-distribution calibration using high-confidence samples $D_H$ obtained in Step 2, while further applying data perturbation to increase robustness. Finally, labeled samples $D_L$, high-value pseudo-labels $D_V$, and distribution-calibrated samples $D_C \in D_H$ are jointly used to perform semi-supervised fine-tuning.

During the final fine-tuning stage, to prevent abrupt changes in the optimization direction and mitigate catastrophic forgetting, we employ a progressive training strategy. The weights of the labeled samples $D_L$ and distribution-calibrated samples $D_C$ are fixed at 1, while the high-value pseudo-labels $D_V$ start with a weight of 0.2 and gradually increased to 1.

### 3.3. Adaptive Fusion Change Decoder

In terms of network architecture and parameter configuration, to reduce the fine-tuning cost, we freeze the encoder and lightweight adaptation via LoRA, while employing the proposed Adaptive Fusion Change Decoder (AFCD) as the task-specific decoder. The network architecture of IAF-CDNet is illustrated in Figure 3. Specifically, the bi-temporal HSIs are first tokenized and fed into the encoder for feature extraction, yielding four hierarchical feature representations from shallow to deep layers, denoted as $\{V_1^1, V_2^1, V_3^1, V_4^1\}$ for the T1 image and $\{V_1^2, V_2^2, V_3^2, V_4^2\}$ for the T2 image. Subsequently, point-wise convolutions are applied to the bi-temporal features, and the resulting features are then fed into AFCD for feature fusion and change information extraction.

In AFCD, a subtraction operation is used to generate initial difference features $\{F_1, F_2, F_3, F_4\}$. Then, AFCD adopts a dual-path progressive fusion strategy, consisting of a semantic path (upper part) and a detail path (lower part). The semantic path propagates global semantic information from deep to shallow layers, thereby enhancing high-level semantic consistency and providing robust contextual guidance for change interpretation. The semantic path can be formulated as:

$$
\begin{aligned}
F_4^S &= F_4, \\
F_{i-1}^S &= CFB([F_i^S, F_{i-1}]), \quad i = 1, 2, 3
\end{aligned}
\tag{4}
$$

where $[\cdot]$ denotes channel-wise concatenation and $CFB(\cdot)$ represents the convolutional fusion block. In contrast, the detail path aggregates local change cues from shallow to deep layers, enabling precise modeling of fine-grained and spatially localized variations. The detail path can be formulated as:

$$
\begin{aligned}
F_1^D &= F_1, \\
F_{i+1}^D &= CFB([F_i^D, F_{i+1}]), \quad i = 1, 2, 3
\end{aligned}
\tag{5}
$$

To balance semantic stability and fine-grained changes, gated fusion blocks are introduced to adaptively fuse two paths:

$$
F_i^G = \begin{cases} GFB(F_3^S, F_2^D), & i = 1 \\ GFB(F_{i-1}^G, LN([F_{4-i}^S, F_{i+1}^D])), & i = 2, 3 \end{cases}
\tag{6}
$$

where $GFB(\cdot)$ denotes the gated fusion block. Finally, the refined feature $F_3^G$ is fed into a lightweight classification head composed of convolutional and activation layers to generate the final change map.

## 4. Experiments

Here, we explain the implementation details and conduct experiments on three benchmark datasets. Moreover, we conduct value analysis of pseudo-labels as well as the transferability and effectiveness evaluation of IA-SFT. Further analysis experiments on IA-SFT components are shown in Appendix C-F.

### 4.1. Implementational details

All experiments are conducted on a single RTX 5090 GPU. The patch size of the tokens and the batch size are set to $5 \times 5$ and 256. In the fine-tuning process, AdamW optimizer is employed with a cosine annealing learning rate schedule, where the initial learning rate is set to 5e-4 and gradually decays to 1e-4 over 100 epochs; In Step 2, the confidence interval for selecting high-confidence samples is set to 90% - 95%; the intervals for selecting low-confidence samples are set to 5% - 10% on River, and 2% - 7% on other datasets; In Step 3, the rapid fine-tuning iterations is set to 3. AdamW

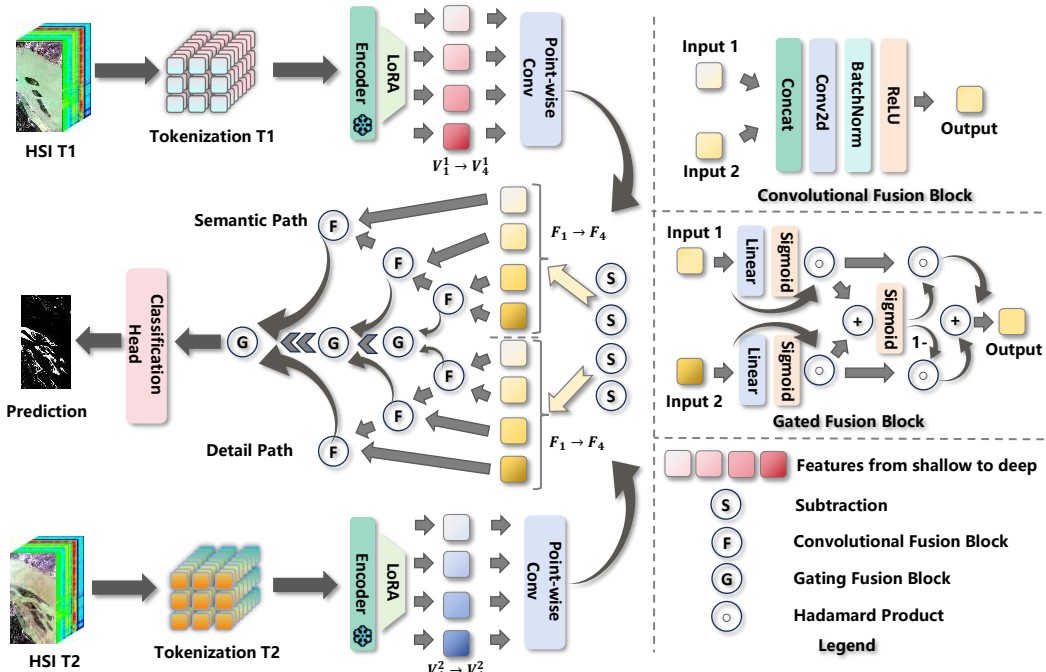

*Figure 3.* Network architecture of IAF-CDNet.

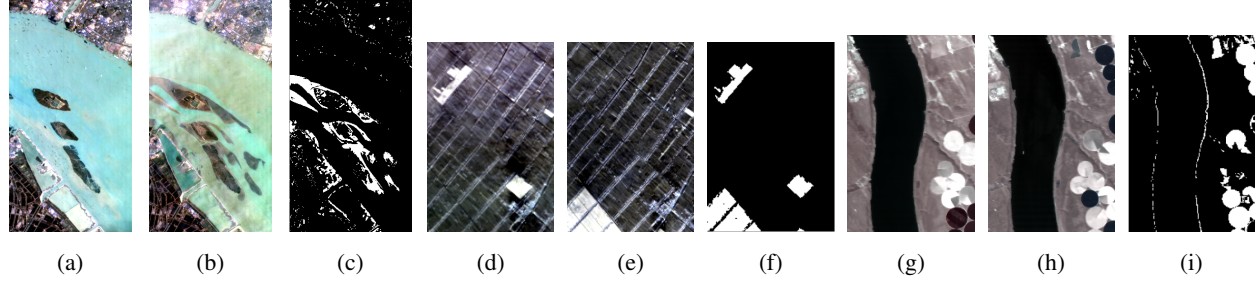

|(a)|(b)|(c)|(d)|(e)|(f)|(g)|(h)|(i)|

*Figure 4.* Visualization of three datasets. (a) River imagery on May 3, 2013. (b) River imagery on Dec 31, 2013. (c) Ground truth of River. (d) China imagery on May 3, 2006. (e) China imagery on April 23, 2007. (f) Ground truth of China.(g) USA imagery on May 1, 2004. (h) USA imagery on May 8, 2007. (i) Ground truth of USA.

optimizer is employed with a learning rate of 1e-6; In Step 4, the maximum sensitivity hyperparameter $b_h$ is set to 0.1. In Step 5, high-value pseudo-labels are selected from those whose mean and variance of influence scores fall within the top 30%.

### 4.2. Datasets

The used datasets include three benchmark HSI-CD datasets, namely River (Wang et al., 2018), China (Hasanlou & Seydi, 2018), and USA (Hasanlou & Seydi, 2018). The visualization of HSIs and ground truth are presented in Figure 4. To increase the challenge of USA and China, the HSIs are cropped to simulate severe class imbalance. Following common practice in HSI-CD, we adopt a transductive semi-supervised setting, where a small set of labeled pixels

(1% for training and 1% for validation) and a large set of unlabeled pixels (98% for testing) are sampled. Detailed dataset information is provided in Appendix B.

### 4.3. Comparative Investigations

Comparison methods include both supervised and semi-supervised methods: DIEFEN (Wu et al., 2024), DCENet (Luo et al., 2024), GlobalMind (Hu et al., 2024), MSCSCNet (Qu et al., 2024), CSCANet(Zhang et al., 2025), MS2FN (Gao et al., 2025), AIWSEN (Wu et al., 2025), PUL-DCD (Zhao et al., 2025), SpikeHCD (Mei et al., 2025). All hyperparameters of the comparison methods are tuned on the validation set.

The CD results are illustrated in Figures 5-7, where True Positive, True Negative, False Positive and False Negative

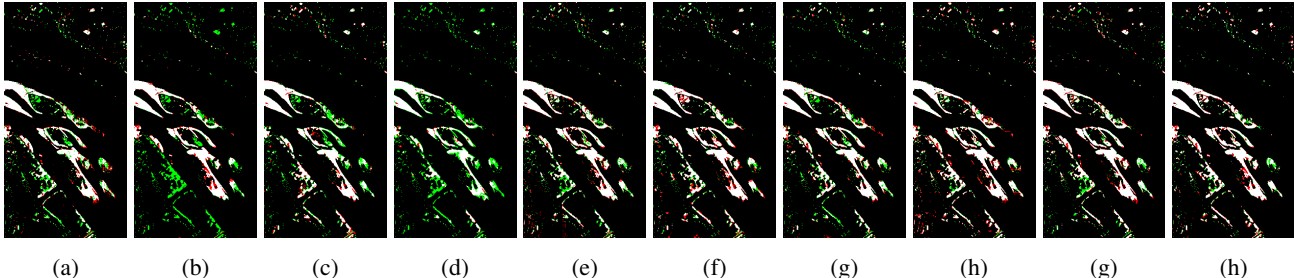

|  (a)  |  (b)  |  (c)  |  (d)  |  (e)  |  (f)  |  (g)  |  (h)  |  (g)  |  (h)  |

*Figure 5.* Change detection results on River dataset. (a) DIEFEN. (b) DCENet. (c) GlobalMind. (d) MSCSCNet. (e) CSCANet. (f) MS2FN. (g) AIWSEN. (h) PUL-DCD. (i) SpikeHCD. (j) IAF-CDNet.

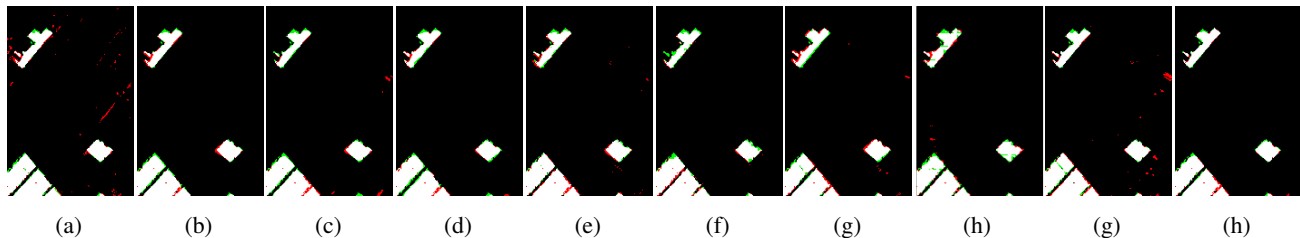

|  (a)  |  (b)  |  (c)  |  (d)  |  (e)  |  (f)  |  (g)  |  (h)  |  (g)  |  (h)  |

*Figure 6.* Change detection results on China dataset. (a) DIEFEN. (b) DCENet. (c) GlobalMind. (d) MSCSCNet. (e) CSCANet. (f) MS2FN. (g) AIWSEN. (h) PUL-DCD. (i) SpikeHCD. (j) IAF-CDNet.

*Table 1.* Quantitative metrics on River dataset

| Methods | OA | KC | Pre | Rec | F1 |
|---|---|---|---|---|---|
| DIEFEN | 96.03 | 73.58 | 80.73 | 71.32 | 75.74 |
| DCENet | 95.58 | 68.00 | 84.50 | 60.21 | 70.32 |
| GlobalMind | 96.42 | 75.72 | 84.80 | 71.62 | 77.65 |
| MSCSCNet | 95.48 | 64.73 | **91.78** | 52.71 | 66.96 |
| CSCANet | 96.49 | 77.43 | 81.22 | 77.55 | 79.34 |
| MS2FN | 96.86 | 80.38 | 81.35 | 82.86 | 82.10 |
| AIWSEN | 96.56 | 77.41 | 83.37 | 75.57 | 79.28 |
| PUL-DCD | 96.30 | 80.45 | 78.54 | 86.09 | 82.14 |
| SpikeHCD | 96.87 | 80.16 | 82.43 | 81.32 | 81.87 |
| IAF-CDNet | **97.28** | **83.11** | 83.14 | **86.11** | **84.61** |

*Table 2.* Quantitative metrics on China dataset

| Methods | OA | KC | Pre | Rec | F1 |
|---|---|---|---|---|---|
| DIEFEN | 98.41 | 90.07 | 88.10 | 93.07 | 90.94 |
| DCENet | 98.75 | 91.97 | 93.44 | 91.88 | 92.65 |
| GlobalMind | 98.52 | 90.47 | 92.73 | 89.86 | 91.27 |
| MSCSCNet | 98.59 | 90.80 | **94.03** | 89.23 | 91.67 |
| CSCANet | 98.55 | 90.94 | 90.19 | 93.31 | 91.73 |
| MS2FN | 98.62 | 91.02 | 93.89 | 89.76 | 91.78 |
| AIWSEN | 98.30 | 89.45 | 88.10 | 92.76 | 90.37 |
| PUL-DCD | 98.43 | 90.85 | 93.38 | 90.11 | 91.72 |
| SpikeHCD | 98.35 | 89.83 | 87.72 | 93.97 | 90.74 |
| IAF-CDNet | **98.95** | **93.41** | 92.09 | **95.96** | **93.99** |

*Table 3.* Quantitative metrics on USA dataset

| Methods | OA | KC | Pre | Rec | F1 |
|---|---|---|---|---|---|
| DIEFEN | 95.72 | 76.01 | 85.67 | 72.22 | 76.01 |
| DCENet | 96.61 | 81.60 | 87.72 | 79.62 | 83.48 |
| GlobalMind | 95.71 | 78.22 | 78.28 | 83.13 | 80.63 |
| MSCSCNet | 95.85 | 77.21 | 84.72 | 74.91 | 79.51 |
| CSCANet | 96.42 | 80.32 | **88.19** | 77.15 | 82.30 |
| MS2FN | 96.35 | 80.01 | 87.20 | 77.44 | 82.03 |
| AIWSEN | 95.81 | 77.96 | 81.11 | 79.52 | 80.31 |
| PUL-DCD | 96.55 | 80.38 | 87.36 | 77.76 | 82.28 |
| SpikeHCD | 96.18 | 78.85 | 87.22 | 75.54 | 80.96 |
| IAF-CDNet | **97.04** | **84.41** | 87.13 | **85.02** | **86.06** |

are represented by white, black, red, and green, respectively. The corresponding quantitative evaluation results are reported in Tables 1-3, where commonly used metrics are presented, including Overall Accuracy (OA), Kappa coefficient (KC), Precision (Pre), Recall (Rec), and F1-score (F1).

On all datasets, IAF-CDNet consistently shows superior performance in both qualitative and quantitative evaluations. Visual comparisons reveal that many existing methods suffer from severe missed detections or excessive false alarms, reflecting an imbalance between precision and recall. In contrast, IAF-CDNet effectively suppressing pseudo-changes while accurately capturing complex and fine-grained change patterns. These observations are well supported by quantitative results, where IAF-CDNet attains the highest KC and F1

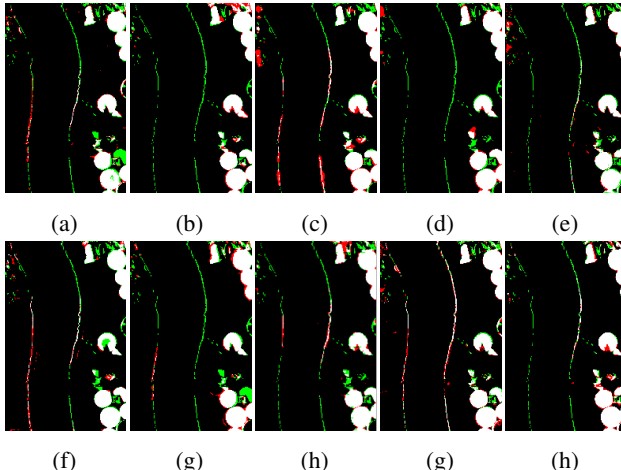

*Figure 7.* Change detection results on USA dataset. (a) DIEFEN. (b) DCENet. (c) GlobalMind. (d) MSCSCNet. (e) CSCANet. (f) MS2FN. (g) AIWSEN. (h) PUL-DCD. (i) SpikeHCD. (j) IAF-CDNet.

*Table 4.* Ablation study of key components

| Fine-tuning | IA-SFT | AFCD | OA | KC | Pre | Rec | F1 |
|---|---|---|---|---|---|---|---|
| Full FT | × | × | 95.67 | 72.97 | 73.91 | 76.84 | 75.35 |
| LoRA FT | × | × | 96.52 | 76.64 | 84.72 | 73.16 | 78.52 |
| LoRA FT | ✓ | × | 96.97 | 81.31 | 81.06 | 84.99 | 82.98 |
| LoRA FT | × | ✓ | 97.21 | 80.77 | **92.05** | 74.35 | 82.26 |
| LoRA FT | ✓ | ✓ | **97.28** | **83.11** | 83.14 | **86.11** | **84.61** |

scores on all three datasets, outperforming the second-best methods by clear margins demonstrating strong robustness to diverse change characteristics.

### 4.4. Ablation Study

To further validate the effectiveness of the key components in IAF-CDNet, ablation experiments are conducted on River dataset.

**Component Effectiveness:** The effectiveness of key components is validated by changing the fine-tuning strategy and removing key modules (Table 4). Due to the limited number of labeled samples, LoRA-based fine-tuning (LoRA FT) achieves better performance than full-parameter fine-tuning (Full FT). From baseline with LoRA FT, the introduction of IA-SFT and AFCD individually leads to improvements of 4.67% and 4.13% in KC, respectively, demonstrating the effectiveness of both components. When both modules are applied jointly, the model achieves the best performance, resulting in an improvement of 6.47% in KC compared to baseline with LoRA FT.

**Impact of IA-SFT on Pseudo-Labels:** We further investigate the impact of IA-SFT by varying the percentile $\delta\%$ used to select high-value pseudo-labels based on their mean and variance (denoted as selection ratio). As illustrated in

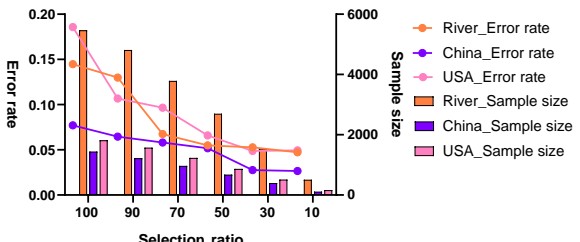

*Figure 8.* Impact of IA-SFT on pseudo-labels

*Table 5.* Comparison of pseudo-label strategies

| Method | River | | China | | USA | |
|---|---|---|---|---|---|---|
| | PLA | F1 | PLA | F1 | PLA | F1 |
| FixMatch | 86.52 | 81.94 | 94.33 | 92.90 | 82.09 | 83.72 |
| FlexMatch | 85.83 | 81.84 | 94.38 | 92.44 | 81.89 | 84.22 |
| SemiReward | 88.63 | 82.87 | 96.73 | 93.50 | 80.35 | 83.14 |
| CGMatch | 86.37 | 82.03 | 94.50 | 92.66 | 82.44 | 84.07 |
| ISAL | 92.31 | 83.53 | 95.83 | 92.60 | 82.36 | 83.55 |
| eFUMI | 76.75 | 82.18 | **97.92** | 92.95 | 88.94 | 83.92 |
| IA-SFT | **94.73** | **84.61** | 97.24 | **93.99** | **95.12** | **86.06** |

Fig. 8, the line plot depicts the pseudo-label error rate, while the bar chart shows the number of selected samples. As $\delta$ decreases, the pseudo-label error rate drops significantly, accompanied by an approximately linear reduction in sample size. This trend indicates a strong correlation between sample quality and the mean–variance criteria, which further validates the effectiveness of IA-SFT.

### 4.5. Comparison of Pseudo-Label Strategies

We further conduct a comparative study of different pseudo-label filtering strategies, including FixMatch (Sohn et al., 2020), FlexMatch (Zhang et al., 2021), SemiReward (Li et al., 2023), CGMatch (Cheng et al., 2025), ISAL (Liu et al., 2021), and eFUMI (Zou & Zare, 2016). For a fair comparison, all methods adopt the same backbone IAF-CDNet, and operate on the same unlabeled set composed of low-confidence samples. As reported in Table 5, FixMatch, FlexMatch and CGMatch yield only marginal improvements in pseudo-label accuracy (PLA), which leads to suboptimal semi-supervised performance. SemiReward and ISAL achieves noticeable PLA gains on the River and China datasets but fails to on USA dataset. Although eFUMI significantly enhances the PLA on China and USA, it consistently tends to favor majority class samples while neglecting valuable minority class samples, thereby yielding limited improvements in the detection performance. In contrast, IA-SFT consistently delivers the best pseudo-label filtering performance on all datasets.

*Table 6.* Evaluation of the generality of the proposed IA-SFT

| Methods | Dataset | Mode | PLA | KC | F1 |
|---|---|---|---|---|---|
| AIWSEN | River | w/o IA-SFT | 86.28 | 77.41 | 79.28 |
| | | w/ IA-SFT | **90.71** | **78.57** | **80.36** |
| | China | w/o IA-SFT | 90.97 | 89.45 | 90.37 |
| | | w/ IA-SFT | **97.31** | **90.72** | **91.52** |
| | USA | w/o IA-SFT | 89.33 | 77.96 | 80.31 |
| | | w/ IA-SFT | **95.81** | **80.39** | **82.32** |
| SpikeHCD | River | w/o IA-SFT | 90.78 | 80.16 | 81.87 |
| | | w/ IA-SFT | **95.86** | **81.23** | **82.75** |
| | China | w/o IA-SFT | 93.51 | 89.83 | 90.74 |
| | | w/ IA-SFT | **98.23** | **92.11** | **92.77** |
| | USA | w/o IA-SFT | 85.39 | 78.85 | 80.96 |
| | | w/ IA-SFT | **94.74** | **80.15** | **82.05** |

### 4.6. Transferability Evaluation

To verify the transferability of IA-SFT, we integrate it into two compared methods, AIWSEN and SpikeHCD, in a plug-and-play manner. As reported in Table 6, without IA-SFT, the PLA is entirely determined by the performance of the initial model and remains relatively low. After applying IA-SFT, PLA is significantly improved, and incorporating the refined pseudo-labels into semi-supervised training leads to notable performance gains for both methods.

## 5. Conclusion

In this paper, we investigate the challenge of reliably leveraging unlabeled data for HSI-CD. We argue that previous methods are limited by confidence-driven pseudo-label selection, which either underutilizes informative samples or introduces harmful noise. To address this issue, we proposed Influence-Aware Semi-supervised Fine-tuning, a novel framework that explicitly measures the influence of pseudo-labels on model predictions over labeled data. IA-SFT enables effective identification of high-value pseudo-labels and improves pseudo-label quality. In addition, we design an Adaptive Fusion Change Decoder to better adapt pre-trained foundation models to HSI-CD. Extensive experiments on multiple benchmark datasets demonstrate that the proposed method outperforms state-of-the-art methods. Further analyses confirm the robustness of IA-SFT and its strong generalization capability when integrated into different frameworks.

## Acknowledgements

This research has been supported by the National Natural Science Foundation of China (62271400). The authors would like to thank the three anonymous reviewers for their insightful comments.

## Impact Statement

This paper presents work whose goal is to advance the field of Machine Learning. There are many potential societal consequences of our work, none which we feel must be specifically highlighted here.

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

## A. Limitations and Challenges

Despite the promising performance demonstrated by the proposed IAF-CDNet, several limitations and challenges remain and warrant further investigation. The most notable limitation lies in the computational cost of the sample-wise rapid fine-tuning stage. Although the encoder is frozen to reduce overhead, the influence evaluation still requires multiple forward and backward passes, which may introduce non-negligible time consumption. A promising direction is to evaluate influence at the group or mini-batch level rather than strictly on a per-sample basis. Pseudo-labels with similar confidence and spatial characteristics can be clustered and jointly fine-tuned, allowing influence scores to be computed in a batch-wise manner. The appropriate optimization strategies remain to be explored further.

The effectiveness of IA-SFT depends on several hyperparameters, such as the seletion ratio $\delta$, the boundary-aware weighting parameters, and the number of rapid fine-tuning iterations. While empirical studies show that the method is relatively robust within a reasonable range, suboptimal parameter choices may lead to overly conservative or overly aggressive pseudo-label selection. Designing adaptive or data-driven parameter tuning mechanisms remains an open challenge.

Similar to most existing semi-supervised HSI-CD methods, the proposed approach operates under a transductive setting, where unlabeled samples are drawn from the test set. Although this setting is widely adopted in HSI-CD due to data constraints, extending the method to inductive scenarios remains an important direction for future work.

## B. Dataset Information

To assess the performance of the proposed IAF-CDNet, experiments are conducted on three widely used HSI-CD datasets, namely River, USA, and China. These datasets were collected by the Earth Observing-1 Hyperion sensor, which operates over a spectral range from 0.4 to 2.5 $\mu$m and provides 242 spectral bands with a spectral resolution of 10 nm and a spatial resolution of 30 m. Despite the large number of spectral bands, hyperspectral data quality is often influenced by atmospheric effects and other environmental factors. As a result, prior studies typically retain only bands with relatively high signal-to-noise ratios for analysis. For completeness, detailed descriptions of the datasets are provided below.

River dataset: This dataset comprises a pair of hyperspectral images acquired on May 3, 2013, and December 31, 2013, over Jiangsu Province, China. The images have a spatial size of $463 \times 241$ pixels and contain 198 spectral bands. The dominant change pattern corresponds to the disappearance of materials within the river area. According to the ground-truth annotations, the dataset includes 9,698 changed pixels and 101,885 unchanged pixels, leading to a pronounced class imbalance that poses substantial challenges for accurate change detection.

USA dataset: The USA dataset consists of bi-temporal HSIs captured on May 1, 2004, and May 8, 2007, in Hermiston, USA. Each image has a spatial resolution of $307 \times 241$ pixels with 154 spectral bands. The scenes mainly cover bare farmland, irrigated farmland, and river regions. To increase the challenge of this dataset, half of the image is selected to emulate a change detection scenario with severe class imbalance. The cropped region has a spatial size of $154 \times 241$ pixels, containing 3,989 changed pixels and 33,125 unchanged pixels.

China dataset: This dataset includes bi-temporal HSIs acquired on May 3, 2006, and April 23, 2007, in Yancheng, Jiangsu Province, China. The original images have a spatial size of $420 \times 140$ pixels and comprise 154 spectral bands. The primary changes are related to farmland areas, with change regions appearing relatively concentrated in the ground-truth map. To simulate a highly imbalanced change detection setting, half of the image is selected. The resulting subset has a spatial resolution of $210 \times 140$ pixels and contains 2,525 changed pixels and 26,875 unchanged pixels.

## C. Effect of boundary- and inconsistency-aware weighting

We also conduct an ablation study of IA-SFT by removing boundary-aware weighting (BAW) and inconsistency-aware weighting (IAW). As shown in Table 7, incorporating BAW improves the pseudo-label accuracy (PLA) on all datasets, with gains of 0.98%, 0.55%, and 1.28% on River, China, and USA datasets, respectively. Compared with BAW, the integration of IAW alone did not yield a significant improvement in PLA. Compared to baseline, joint application of both strategies results in overall improvements of 1.22%, 0.74%, and 1.55%. In general, the ablation results validate the effectiveness and necessity of both weighting strategies.

*Table 7.* Ablation study of boundary- and inconsistency-aware weighting

| Prediction Shift | BAW | IAW | River PLA | China PLA | USA PLA |
|:---:|:---:|:---:|:---:|:---:|:---:|
| ✓ | ✗ | ✗ | 93.51 | 96.50 | 93.57 |
| ✓ | ✓ | ✗ | 94.49 | 97.05 | 94.85 |
| ✓ | ✗ | ✓ | 93.92 | 96.69 | 94.08 |
| ✓ | ✓ | ✓ | **94.73** | **97.24** | **95.12** |

*Table 8.* Influence of Rapid Fine-Tuning Iterations in IA-SFT on Pseudo-Label Accuracy

| Number of iterations | River PLA | China PLA | USA PLA |
|:---:|:---:|:---:|:---:|
| 0 | 85.63 | 92.78 | 81.87 |
| 1 | 94.47 | 96.42 | **95.86** |
| 2 | 94.65 | 97.15 | 95.45 |
| 3 | 94.73 | 97.24 | 95.12 |
| 4 | 93.95 | **97.59** | 94.58 |
| 5 | **95.38** | 96.96 | 93.57 |

## D. Impact of rapid fine-tuning iterations

Table 8 presents the impact of the number of rapid fine-tuning iterations on pseudo-label accuracy (PLA). When the number of iterations is set to 0, it indicates that IA-SFT filtering is not performed. On the River dataset, the highest PLA is achieved when the number of iterations is set to 5. On the China dataset, PLA consistently exceeds 97% when the iteration number ranges from 2 to 4. In contrast, on the USA dataset, fewer fine-tuning iterations lead to better PLA performance. Overall, when the iteration number is set between 1 and 5, PLA is substantially improved compared to the case without sample filtering, while the performance variation across different iteration numbers remains relatively small. These results demonstrate that IA-SFT is robust to the choice of fine-tuning iterations.

## E. Analysis of the influence evaluation metrics

To investigate the effectiveness of different impact evaluation metrics for pseudo-label selection, we conduct comparative experiments using four statistical criteria, namely mean, variance, maximum (max), and minimum (min) values. The results are summarized in Table 9, where the reported accuracy on each dataset reflects the correctness of pseudo labels selected under different strategies.

When considering individual strategies, both the mean-based and variance-based selection achieve relatively stable performance. Specifically, the mean strategy performs well on the River and USA datasets, while the variance strategy shows advantages on the China dataset. This indicates that the mean captures the overall impact magnitude of pseudo samples, whereas the variance helps characterize prediction stability. However, relying on a single statistic is insufficient to comprehensively assess pseudo-label quality.

In contrast, strategies based on extreme values exhibit noticeably inferior and unstable performance. In particular, the

*Table 9.* The impact of different strategy for selecting pseudo-label

| Strategy | | | | River | China | USA |
|:---:|:---:|:---:|:---:|:---:|:---:|:---:|
| Mean | Variance | Max | Min | | | |
| ✓ | ✗ | ✗ | ✗ | **0.9506** | 0.9492 | 0.9469 |
| ✗ | ✓ | ✗ | ✗ | 0.9488 | 0.9654 | 0.9396 |
| ✗ | ✗ | ✓ | ✗ | 0.9305 | 0.9700 | 0.9359 |
| ✗ | ✗ | ✗ | ✓ | 0.6614 | 0.8799 | 0.6245 |
| ✓ | ✓ | ✓ | ✗ | 0.9399 | 0.9720 | 0.9503 |
| ✓ | ✓ | ✗ | ✗ | 0.9473 | **0.9724** | **0.9512** |

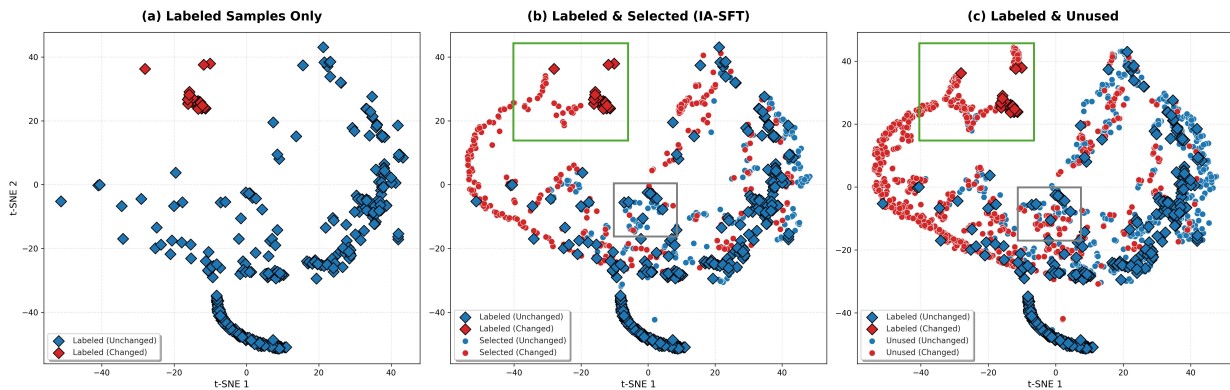

*Figure 9.* Visualization of the pseudo-label selection performance

minimum-based strategy consistently yields the lowest pseudo-label accuracy across all datasets, suggesting that focusing solely on weak or highly unstable responses introduces excessive noise and is unsuitable for pseudo-label selection.

We further evaluate combinations of multiple statistics. The results demonstrate that jointly leveraging the mean and variance leads to the most consistent and competitive performance across all datasets. Introducing additional statistics, such as maximum values, does not provide further benefits and may even degrade performance in certain cases, indicating that excessive emphasis on extreme responses can adversely affect pseudo-label evaluation.

Based on these observations, we adopt the mean-and-variance-based impact quantification strategy in IA-SFT. This design effectively suppresses noisy pseudo labels while reliably preserving informative low-confidence samples, thereby providing more robust supervision for subsequent semi-supervised fine-tuning.

## F. Feature visualization of IA-SFT

We perform an t-SNE analysis to visualize the feature space distributions. Figure 9 provides a comprehensive visualization of the pseudo-label selection performance of our proposed IA-SFT. Specifically, Fig. 9(a) illustrates the initial feature space distribution of the labeled samples; Fig. 9(b) shows the feature space distribution of the pseudo-labels selected by IA-SFT and the distribution of labeled samples; Fig. 9(c) shows the distribution of unselected pseudo-labels and labeled samples.

By comparing the regions within the green bounding boxes, it is evident that IA-SFT effectively reduces the probability of selecting low-information redundant samples and successfully filters out noisy candidates. Furthermore, a comparison of the gray bounding boxes, which represent the hard-to-classify regions, reveals that IA-SFT strictly limits the selection of pseudo-labels within these ambiguous zones. IA-SFT maintains a relatively clear inter-class boundary and prevents the propagation of errors caused by a large number of mislabeled samples. These visualization results clearly demonstrate the effectiveness of IA-SFT in selecting high-value and reliable pseudo-labels.

To further demonstrate the performance gain, we provide a comparison of the feature representations on the test set before and after applying the IA-SFT strategy, as shown in Fig. 10. This visualization highlights the improvement of the model's discriminative power.

Comparing Fig. 10(a) and Fig. 10(b), there is a marked improvement in the separation of class clusters. Before IA-SFT, the features of changed (red) and unchanged (blue) samples are relatively dispersed, with noticeable overlap at the decision boundary. After IA-SFT fine-tuning, the two classes are mapped into more distinct and isolated manifolds. Additionally, it can be observed that the number of misclassified samples is also significantly reduced in regions far from the decision boundary. Thus, Fig. 10 demonstrates the effectiveness of IA-SFT-selected pseudo-labels in enhancing the model's discriminative capability.

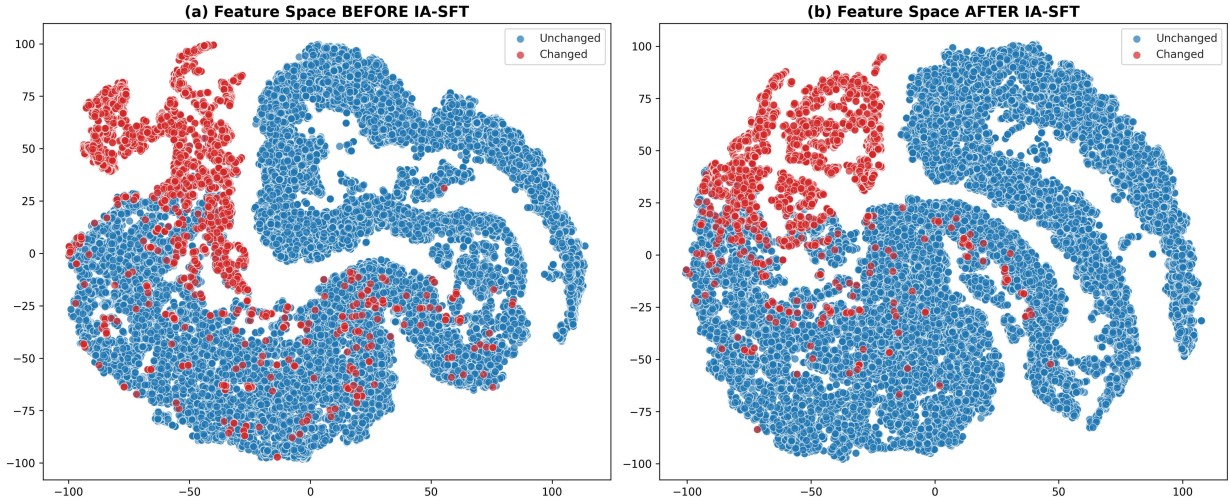

*Figure 10.* Visualization of the improved discriminative ability

