# OpenReview forum: "Rethinking Low-Confidence Pseudo Labels: Influence-Aware Semi-Supervised Fine-Tuning for Hyperspectral Change Detection"
_ICML.cc/2026/Conference — ICML 2026 regular_

### Official Review · Reviewer_TuNp · 2026-03-05

**Soundness:** 4
**Presentation:** 4
**Significance:** 3
**Originality:** 3
**Overall Recommendation:** 4
**Confidence:** 3

**Summary:**

This paper studies hyperspectral image change detection (HSI-CD) under scarce annotations, where fine-tuning tends to overfit. It proposes IA-SFT, a semi-supervised fine-tuning framework that selects low-confidence pseudo-labeled samples by measuring their influence on decisions over labeled data, aiming to keep pseudo labels that are informative while less noisy. The paper further designs an AFCD decoder to jointly model global semantic consistency and local change details. Experiments on three HSI-CD benchmarks show consistent improvements over prior methods across multiple metrics. Overall, the motivation is meaningful, the design is targeted, and the paper is clearly written, while several aspects warrant further discussion.

**Compliance With Llm Reviewing Policy:**

Affirmed.

**Final Justification:**

This rebuttal has addressed most of my concerns and has helped clarify both the strengths and limitations of the paper. The additional experiments on computational overhead and distribution shifts are convincing. However, in my view, the explanation and justification regarding training stability under high-variance samples remain insufficiently detailed. Overall, the paper presents a novel and practically meaningful idea, is technically rigorous, supported by solid experiments, and evaluated quite comprehensively. Therefore, I maintain my original score of 4.

**Key Questions For Authors:**

1. In Step 2, how are the thresholds chosen to remove extremely low-confidence and extremely high-confidence samples? Are these thresholds overly sensitive across different datasets?
2. For the influence computation via rapid fine-tuning: which parameters are updated (LoRA only, decoder only, or both)? How sensitive are results to the number of rapid-FT steps and learning rate? Please include a small sensitivity study.
3. In Section 3.2.4, the final “high-value” pseudo labels are selected as those whose influence-score mean and variance are both among the top group. Since high-variance supervision can lead to unstable optimization, do you observe training instability or oscillations? If so, how do you mitigate it?
4. Please add quantitative compute cost reporting (time and memory), ideally with a stage-wise breakdown, to better support the practicality of IA-SFT.

**Limitations:**

A key limitation is the additional overhead introduced by sample-wise rapid fine-tuning and the reliance on a transductive setting. The scalability and robustness of the approach under larger-scale / higher-resolution unlabeled data and under distribution shift (e.g., cross-region or cross-time deployment) remain to be validated.

**Strengths And Weaknesses:**

Strengths
1. Novel angle on low-confidence pseudo labels. Reconsidering the value of low-confidence pseudo labels is a promising direction. Replacing confidence-based filtering with an influence-based criterion (impact on labeled-data decisions) could be useful not only for remote sensing but also for broader CV settings.
2. Clear presentation. The paper is well organized with high-quality figures. Figure 2 provides a clear multi-stage pipeline overview, making the method easy to follow.

Weaknesses
1. Attribution between IA-SFT and AFCD is not fully isolated. The main novelty lies in influence-based mining of low-confidence pseudo labels, but AFCD adds non-trivial decoder capacity (multi-level two-branch fusion plus gated fusion). Please strengthen controlled ablations to show that IA-SFT yields comparable gains with a simpler decoder / the backbone’s default decoder, and quantify the marginal gain of AFCD alone.
2. Compute overhead may be substantial. The sample-wise (or small-batch-wise) rapid fine-tuning used to estimate influence appears to require additional forward/backward passes per candidate, which could dominate training cost. Please report wall-clock time / GPU-hours and memory, ideally with a stage-wise breakdown (Steps 1–5) and a comparison against standard semi-supervised baselines.
3. Influence proxy may conflate informativeness with instability/noise. A large prediction shift on labeled data after rapid fine-tuning could reflect genuinely informative hard samples, but could also come from noisy pseudo labels or samples that induce harmful gradients. Beyond aggregate performance gains, please provide more direct evidence that high influence-score samples are more likely to be correctly pseudo-labeled (e.g., correlation with pseudo-label accuracy or stratified analysis on boundary regions), and discuss failure modes.
4. The benchmarks are standard for HSI-CD but have relatively small spatial sizes. Sample-wise fine-tuning may be feasible at this scale, but scalability to larger unlabeled pools and higher-resolution remote-sensing scenarios is unclear. Some acceleration strategy would likely be necessary.

---

> ### Author Rebuttal · Authors · 2026-03-31
>
> Thanks for your valuable feedback. Our responses are as follows.
>
> W1: Ablation study of IA-SFT and AFCD
>
> **Response:** In manuscript Table 4, we evaluated IA-SFT and AFCD individually on the LoRA baseline. Using the default backbone decoder, IA-SFT alone improves F1 by 4.4%, while AFCD alone yields 3.7%.
>
> W2&Q4: Computational overhead
>
> **Response:** We performed a stage-wise complexity analysis against a standard dynamic threshold pseudo-label (DT) baseline. DT iteratively selects high-confidence samples for training, adding at most 5%, 20%, and 50% of the unlabeled samples throughout the entire training process. Results are shown below.
>
> **Table 1. Complexity experiments**
>
> |Method|Learnable Params(M)|Peak Memory(GB)|Wall-clock time(s)|
> |--|--|--|--|
> |Step 1|1.74|27|231|
> |Step 2|/|/|0.02|
> |Step 3|0.58|3|114|
> |Step 4|/|/|0.7|
> |Step 5|1.74|27|796|
> |Ours (5)|1.74|27|1142|
> |DT(5)|1.74|27|757|
> |Ours (20)|1.74|27|2668|
> |DT(20)|1.74|27|1865|
> |Ours (50)|1.74|27|3738|
> |DT(50)|1.74|27|3623|
>
> Critically, since this stage only updates the decoder, we can pre-extract features through a single forward pass of the backbone.  Consequently, the overhead of Step 3 is governed by the decoder, ensuring the process is both memory-efficient and computationally tractable. Moreover, the low memory requirement (3GB) allows for high-throughput batch-wise parallelization, further reducing the overall wall-clock time.
>
> W3: Evidence for influence-accuracy correlation
>
> **Response:** Manuscript Fig. 8 shows a consistent trend that increasing the selection threshold (focusing on higher-influence samples) steadily reduces the error rate, suggesting that high-influence samples are of higher quality. Additionally, statistical analysis over batches indicates that the mean influence of correctly labeled samples (4.8e-4) is higher than both the global mean (4.5e-4) and incorrectly labeled samples (3.1e-4). Similar behavior is also observed in terms of variance.
>
> Q1: Threshold sensitivity
>
> **Response:** We use a unified 5% percentile across datasets for adaptive selection. Experiments (will include) show that selecting within a 1–10% range for low-confidence samples yields stable results, with F1 fluctuations <1%. High-confidence samples serve to stabilize the model against intense shifts; thus, within a reasonable range, the threshold has a limited effect on performance.
>
> Q2: Fine-tuning strategy & Hyperparameters
>
> **Response:** We adopt a decoder-only strategy in step 3. Thus, the computational cost comes only from the lightweight decoder. Sensitivity studies in Appendix Table 8 confirm robustness to step numbers. Additional tests (will include) show that small learning rates consistently yield stable convergence.
>
> Q3: Training stability
>
> **Response:** High variance in influence scores is actually a signature of reliable knowledge integration rather than stochastic instability. A high variance indicates that the influence is targeted. This selective impact results in a high variance of influence scores across the training set. In contrast, an incorrect label produces a weak and diffuse influence (low mean), representing gradient interference that conflicts with the model’s established knowledge. Thus, the high variance is not a manifestation of chaotic fluctuations, but a reflection of consistency reinforcement.
>
> Limitation: Robustness to distribution shift and high-resolution scenarios.
>
> **Response:** For distribution shift, supplementary experiments on unseen regions (China/USA) demonstrate robust distribution shift generalization (see response to Reviewer #UKyi)
>
> For high-resolution scenarios, we conducted experiments on the benchmark WHU by applying IA-SFT to the baseline FILFBCD. We selected 10% and 90% of the training set as labeled and unlabeled data, and performed validation on the unseen test set. Results are shown below.
>
> **Table 2. Experiments on WHU**
>
> |Method|Wall-clock time(s)|IoU|Pre |Rec|F1|
> |-|-|-|-|-|-|
> |Baseline|1197|68.8|78.2|85.2|81.5|
> |+IA-SFT|+1606 (247+1359)|71.8|82.5|84.6|83.5|
>
> Table 2 shows that IA-SFT boosts IoU and F1 by 3.0% and 2.0%, respectively. Although it introduces additional overhead (247s), this cost is amortized by filtering 78% of low-confidence samples, which substantially reduces redundant backpropagation. This selective mechanism ensures that while IA-SFT may exhibit modest efficiency gains in small-sample scenarios, it still possesses superior scalability and net efficiency when applied to large-scale unlabeled data.

---

> > ### Author Rebuttal · Reviewer_TuNp · 2026-04-01
> >
> > Thank you for resolving my doubts, so I retained the original positive rating.

---

> > > ### Author Response · Authors · 2026-04-03
> > >
> > > We sincerely thank the reviewer for the valuable comments, which have helped us further improve the paper. We are glad that our responses have addressed the concerns and questions raised.

---

### Official Review · Reviewer_JYZc · 2026-03-12

**Soundness:** 3
**Presentation:** 4
**Significance:** 2
**Originality:** 2
**Overall Recommendation:** 4
**Confidence:** 5

**Summary:**

This paper proposes the concept of selecting points  with pseudo-labels for use within a semi-supervised framework by influence values instead of confidence values.  The approach (IAF-CDNet) focuses on the task of hyperspectral change detection.

**Compliance With Llm Reviewing Policy:**

Affirmed.

**Final Justification:**

I have revised my overall rating based on the authors' rebuttal to my comments and those of the other reviewers.

**Key Questions For Authors:**

- It is unclear to me why only low-confidence points are examined for influence? Shouldn't this be applied to both high- and low- selected samples?

- How was the weighting scheme determined?  It is unclear to me that "close to the decision boundary" is the key factor to consider here.  Intuitively, wouldn't this also be elevating potentially noisy pseudo-labels?

- How are the architecture decisions outlined important for the overall proposed concept?  Or should this be not dependent on architecture?

**Limitations:**

The proposed approach only considers samples with low confidence values for influence estimation.  However, high confidence values are used as well and may also have varying influence.  Why not apply the concept to both sets of points?

The proposed approach defines influence based on some fine tuning steps. How does the fine tuning strategy impact performance and stability of the approach?

**Strengths And Weaknesses:**

Strengths:  The use of influence (as opposed to confidence) focuses learning on "important" regions of the feature space.  In contrast, uncertainty based approaches are likely to focus on regions that lend themselves to overfitting.  The overall concept is sound and well motivated.

Weaknesses:
- The section "Pseudo-label selection strategy" (or earlier in the paper) focuses primarily on why previous methods are not effective.  This is convincing but would be greatly strengthened if an intuitive explanation of why the proposed approach is effective was included. Related, the panel Figure 1c is not fully clear to me.  What specifically does "influence-aware interaction" mean?

-How does the proposed approach compare to the literature for influence-based sample selection?  It is unclear how the proposed approach improves upon the existing literature in this space for this concept?  For example, the influence selection based approaches in active learning (dot: 10.1109/ICCV48922.2021.00914). Similarly, there is at least one hyperspectral paper with the use of "influence" to select samples from training (https://doi.org/10.1117/12.2228154), how does this approach relate to that concept?

---

> ### Author Rebuttal · Authors · 2026-03-31
>
> Thanks for your insightful feedback. Our responses are as follows.
>
> W1: Effectiveness of IA-SFT & Clarity of Fig. 1c
>
> **Response:**  The effectiveness of IA-SFT stems from the distinct influence patterns that correct and noisy pseudo-labels exert on labeled data. Specifically, noisy pseudo-labels tend to produce unstable changes in model predictions during rapid fine-tuning. This is because such samples often conflict with the dominant feature structure learned from labeled data, and their effects do not align with a coherent optimization direction. As a result, their overall influence is suppressed during the brief fine-tuning.
>
> Conversely, correct pseudo-labels, although sometimes assigned low confidence, are usually aligned with the underlying data manifold. When used for fine-tuning, they induce more consistent and structured prediction shifts on labeled data. Moreover, since they interact differently with various samples, their influence exhibits both a higher mean and greater diversity (variance).
>
> Fig. 1c illustrates that IA-SFT can effectively select correct pseudo-labels while filtering out incorrect ones, leading to a more optimized boundary. Influence-aware interaction represents a bidirectional feedback between pseudo-labels and labeled data. A pseudo-label actively changes the model’s perception of labeled samples through fine-tuning. Then, the prediction shift provides the feedback that evaluates the pseudo-label's utility. This feedback determines the weight of pseudo-labels, realizing an influence-based interactive process.
>
> W2: Influence-based methods
>
> **Response:** The improvements of IA-SFT over prior influence-based methods lie in an effective influence quantification strategy (rapid fine-tuning) and a reliable selection mechanism (statistical filtering). These advancements render IA-SFT more robust to distribution shifts and less sensitive to the initial quality of pseudo-labels.
>
> ISAL (doi: 10.1109/ICCV48922.2021.00914) approximates influence via gradient-based sensitivity, which can be affected by inaccurate pseudo-labels, especially in low-confidence regions (error propagation). eFUMI (doi: 10.1117/12.2228154) measures influence from a representation perspective, focusing on feature reconstruction rather than directly assessing the impact on decision boundaries. In contrast, IA-SFT defines influence based on prediction shifts on labeled data, which directly reflects how a pseudo-label affects model decision behavior, providing a more task-relevant criterion for pseudo-label selection.
>
> We provide empirical comparisons with ISAL and eFUMI; the pseudo-label accuracy (PLA) and F1 are as follows:
>
> **Table 1. Comparison with influence-based methods**
>
> |Method|River-PLA|River-F1|China-PLA|China-F1|USA-PLA|USA-F1|
> |-|-|-|-|-|-|-|
> |ISAL|92.3|83.5|95.8|92.6|82.3|83.5|
> |eFUMI|76.7|82.1|97.9|92.9|88.9|83.9|
> |IA-SFT|94.7|84.6|97.2|93.9|95.1|86.0|
>
> Results show that ISAL and eFUMI lack stability. Although eFUMI achieves the highest PLA on China, its sample distribution of unchanged and changed classes is 142:2, ignoring informative minority samples. Thus, its PLA lead fails to improve F1.
>
> Q1 and Limitation 1: Why are only low-confidence points used
>
> **Response:** We focus on low-confidence samples because influence evaluation is most informative in regions of high uncertainty. High-confidence samples are often redundant easy cases with diminishing returns. Conversely, low-confidence regions harbor both destructive noise and high-value hard samples. Focusing IA-SFT here filters noise and extracts critical information at minimal cost, balancing data diversity and computational efficiency.
>
> Q2: Weighting scheme and noise issue
>
> **Response:** We would like to clarify that there is a misunderstanding: in our scheme, "close to the boundary" refers to the affected labeled data. A higher weight is assigned only when a pseudo-label induces prediction shifts on labeled samples near the boundary, which are known to be the most sensitive regions for refining model decisions. We agree that noise can also cause prediction shifts; however, as analyzed in Weakness 1, our mean-and-variance filtering strategy can distinguish and filter out such noise while retaining informative samples.
>
> Q3:  Impact of architecture
>
> **Response:** IA-SFT is inherently architecture-agnostic. This versatility is confirmed by Table 6 in manuscript (diverse architectures) and our response to Reviewer #TuNp's limitations (high-resolution remote sensing model). These results prove that IA-SFT's effectiveness depends on its universal influence-aware strategy rather than any specific architecture.
>
> Limitation 2: Impact of fine-tuning strategy
>
> Response:  We only fine-tune the decoder during this stage. Compared to encoder-decoder fine-tuning, this strategy remains effective and is more efficient. Stable gains are achieved across a few iteration steps (1–5)(Appendix Table 8), and low learning rates (1e-4 to 1e-7) (F1 fluctuation<1%).

---

> > ### Author Rebuttal · Reviewer_JYZc · 2026-04-03
> >
> > Rebuttal does provide some clarity.  I do encourage the authors to revise their submission to ensure the clarification questions are well addressed.  I feel the term "influence" is defined differently across the relevant literature and this paper redefines it yet again.  So ensuring this is clearly stated and how it relates to existing literature and terms would be helpful.

---

> > > ### Author Response · Authors · 2026-04-04
> > >
> > > We sincerely thank the reviewer for the highly constructive suggestion. We agree that "influence" is a broad concept and is therefore defined differently across the literature. Further refining this definition would indeed improve the clarity and rigor of this manuscript. Following your suggestion, we have revised the submission to more clearly distinguish our approach from existing influence-based methods.
> > >
> > > Specifically, we categorize influence-based pseudo-label selection strategies into four types: gradient-based influence, data value influence, feature reconstruction-based representation influence, and prediction shift influence based on rapid fine-tuning.
> > >
> > > (I）Gradient-based influence [1-3]: defines influence as the expected change in model parameters.
> > >
> > > (II) Data value influence [4]: defines a sample's influence based on its marginal contribution to the overall model performance.
> > >
> > > (III) Representation influence [5]: defines influence based on spectral reconstruction and changes to estimated endmembers.
> > >
> > > (IV) Prediction-shift influence (ours): defines influence as the empirical prediction shift on labeled data, measured before and after rapidly fine-tuning the model using pseudo-labels.
> > >
> > > Accordingly, we have added a more detailed description and clear differentiation of influence-based methods in the Related Work and Method section of the manuscript.
> > >
> > > **In Section 2.2:** While confidence-based strategies effectively filter out low-probability noise, they often overlook the actual contribution of a sample to the model’s generalization. To overcome the limitations of static confidence thresholds, recent studies explored influence-based selection strategies. Given its broad definition, influence has been characterized from multiple perspectives in prior work. Generally, it can be categorized into three established paradigms: Gradient-based influence [1-3], which evaluates the expected change in model parameters via gradient inner products; Data value influence [4], which employs game-theoretic tools like Shapley values to assess a sample’s marginal contribution to overall accuracy; and Representation influence [5], which, particularly in remote sensing, measures how a sample alters spectral reconstruction or endmember estimation. However, the reliability of these influence estimations is often heavily contingent upon the model’s initial performance and exhibits significant instability in low-confidence regions, where volatile gradients may lead to suboptimal sample selection.
> > >
> > > **In Section 3.1:** In this work, we introduce the concept of prediction-shift influence to quantify sample importance. Distinct from the gradient-based or representation-based paradigms discussed in Section 2.2, we define influence as the empirical shift in predictions on labeled data. By employing a rapid, lightweight fine-tuning, IA-SFT identifies candidates that actively steer the decision boundary toward a more generalized state.
> > >
> > > Additionally, we have incorporated quantitative comparisons between IA-SFT and other influence-based methods into Table 5 of the revised manuscript. The results confirm that our task-specific definition of influence leads to more stable and accurate sample selection. We believe that these revisions will further enhance the clarity of the concept of influence.
> > >
> > > [1] Understanding black-box predictions via influence functions. In International conference on machine learning.
> > >
> > > [2] Influence selection for active learning. In Proceedings of the IEEE/CVF international conference on computer vision.
> > >
> > > [3] Estimating training data influence by tracing gradient descent. In Advances in Neural Information Processing Systems.
> > >
> > > [4] Data shapley: Equitable valuation of data for machine learning. In International conference on machine learning.
> > >
> > > [5] Instance influence estimation for hyperspectral target signature characterization using extended functions of multiple instances. In Algorithms and Technologies for Multispectral, Hyperspectral, and Ultraspectral Imagery.

---

### Official Review · Reviewer_UKyi · 2026-03-13

**Soundness:** 4
**Presentation:** 4
**Significance:** 3
**Originality:** 3
**Overall Recommendation:** 5
**Confidence:** 5

**Summary:**

This paper proposes an Inffuence Aware Semi-supervised Fine-tuning (IA-SFT) for hyperspectral change detection. Firstly, the inffuence of pseudo-labels on model decision behavior is evaluated for supervision signals, instead of conffdence-based selection. To further adapt foundation models to hyperspectral image change detection, this paper designs an Adaptive Fusion Change Decoder (AFCD) that jointly models global semantic consistency and local change details. The experimental resutls demonstrate the effectiveness of proposed method.

**Compliance With Llm Reviewing Policy:**

Affirmed.

**Final Justification:**

There are several strengths of this paper as follows:
1. The motivation of this paper is clear, which is designed to measure the sample influence on the model prediction for semi-supervised learning.
2. The design of formulated influence score for each pseudo sample is of soundness, which considers not only the predicted probability but also the label changes.
3. The experimental results are significant, where the proposed IAF-CDNet outperforms compared methods.
However, the feature visualization is still absent in the rebuttal, where more insight supports for the model design and discussion are preferred. I decide to keep my recommendation as accept.

**Key Questions For Authors:**

1. Could the authors clearly articulate the originality and key contributions of this work, as this has not been explicitly stated in the current manuscript?

2. Could the authors provide feature visualization results to better support the analysis and discussion presented in the paper?

3. Could the authors present more comprehensive experimental results and ensure that the inputs are properly aligned with the results shown in Fig. 6 and Fig. 7?

**Limitations:**

The paper should discuss its limitation of not being evaluated on semantic change detection tasks.

**Strengths And Weaknesses:**

Strength:
1. The motivation of this paper is clear, which is designed to measure the sample influence on the model prediction for semi-supervised learning.
2. The design of formulated influence score for each pseudo sample is of soundness, which considers not only the predicted probability but also the label changes.
3. The experimental results are significant, where the proposed IAF-CDNet outperforms compared methods.

Weakness:
1. The originality of this paper has not been stated clearly.
2. The feature visualzation has not been provided to support the discussion of this paper.
3. The displayed experimental results can be more sufficient, where the inputs should be aligned with results in Fig.6 and Fig.7.

---

> ### Author Rebuttal · Authors · 2026-03-31
>
> Thanks for your constructive feedback. Our responses are as follows.
>
> W1&Q1: Clarity of the originality and key contributions
>
> **Response:** Thank you for highlighting this. We have revised the manuscript to explicitly articulate our originality. Specifically, our key contributions are summarized as follows:
>
> (i) Influence-Aware Semi-supervised Fine-Tuning (IA-SFT): We propose a pluggable pseudo-label selection strategy that goes beyond confidence-based selection. By evaluating the influence of pseudo-labels on the model's decision making, IA-SFT enables a more reliable selection of high-value unlabeled samples.
>
> (ii) Adaptive Fusion Change Decoder (AFCD): We design an Adaptive Fusion Change Decoder that integrates global semantic context and local detail features through multi-level feature interactions, enabling more precise modeling of fine-grained change regions.
>
> (iii) Extensive Evaluation and Discussion: We conduct comprehensive experiments across diverse models and benchmarks. The results demonstrate that IA-SFT effectively selects high-quality pseudo-labels, leading to substantial performance improvements in change detection, while maintaining strong scalability and robustness.
>
> W2&Q2: Provide the feature visualization
>
> **Response:** Thank you for this suggestion. To demonstrate that the selected pseudo-labels exhibit feature space distributions more consistent with those of the true labels, we extract features of the unlabeled samples and categorize them based on true label class, pseudo-label class, and whether they are selected. Moreover, by visualizing the features of hard samples extracted by the model before and after fine-tuning, we analyze the impact of pseudo-labels selected by IA-SFT on the decision boundary. The visualizations will be added to the appendix.
>
> W3&Q3: Present more comprehensive experimental results
>
> **Response:** We appreciate this insightful comment. While keeping the available labeled and unlabeled samples consistent with those in Figs. 6 and 7, we used the model to make predictions. The objective metrics are shown below.
>
> **Table 1. Objective metrics on the China dataset**
>
> |Method|OA|KC|Pre|Rec|F1|
> |--|--|--|--|--|--|
> |MS2FN|72.6|46.2|85.1|59.7|70.2|
> |AIWSEN|72.7|46.7|88.6|56.7|69.1|
> |PUL-DCD|73.8|47.1|71.2|83.2|76.8|
> |SpikeHCD|71.8|42.5|69.6|**85.4**|76.7|
> |**Ours**|**87.3**|**73.2**|**89.6**|85.0|**87.3**|
>
> **Table 2. Objective metrics on the USA dataset**
>
> |Method|OA|KC|Pre|Rec|F1|
> |--|--|--|--|--|--|
> |MS2FN|78.0|49.9|70.6|62.1|66.1|
> |AIWSEN|66.2|36.9|55.5|90.2|64.8|
> |PUL-DCD|88.3|72.5|87.4|75.3|80.9|
> |SpikeHCD|83.8|66.7|70.0|**92.5**|79.7|
> |**Ours**|**91.5**|**80.5**|**93.4**|80.9|**86.7**|
>
> This experiment follows the standard inductive semi-supervised learning setup, and the available data exhibit a severe distribution shift from the test set. Thus, most models are prone to significant performance degradation. According to the results, our method achieves the best performance, with advantages that are more pronounced than those shown in Tables 2 and 3 of the manuscript, demonstrating its generalization capability.
>
> Limitations: limitations regarding semantic change detection
>
> **Response:** We appreciate this insightful comment. While the current work focuses on binary change detection, extending IA-SFT to semantic change detection is indeed a promising and feasible direction.
>
> In terms of methodological feasibility, since IA-SFT is fundamentally architecture-agnostic and grounded in prediction shifts, it can be naturally generalized to semantic change detection. In a multi-class setting, instead of a binary shift, IA-SFT would quantify the influence of a pseudo-label on the model’s class-wise probability distributions across labeled data. Furthermore, this could benefit from semantic-similarity-aware influence quantification and more fine-grained influence scoring mechanisms. We will add a dedicated discussion on these future extensions in the revised manuscript to provide a more comprehensive perspective.

---

> > ### Author Rebuttal · Reviewer_UKyi · 2026-04-04
> >
> > I thank the authors for their comprehensive rebuttal. They have thoroughly answered my questions regarding feature visualization, motivation and originality. However, the feature visualization is still absent, where more insight supports for the model design and discussion are preferred. I decide to keep my recommendation as accept.

---

> > > ### Author Response · Authors · 2026-04-04
> > >
> > > Thank you for this constructive suggestion. Following your suggestion, we have performed an extensive t-SNE analysis to visualize the feature space distributions. The results can be obtained at https://anonymous.4open.science/r/TR65Y-D2D1.
> > >
> > > Figure 1 provides a comprehensive visualization of the pseudo-label selection performance of our proposed IA-SFT. Specifically, Fig. 1(a) illustrates the initial feature space distribution of the labeled samples; Fig. 1(b) shows the feature space distribution of the pseudo-labels selected by IA-SFT and the distribution of labeled samples; Fig. 1(c) shows the distribution of unselected pseudo-labels and labeled samples.
> > >
> > > By comparing the regions within the green bounding boxes, it is evident that IA-SFT effectively reduces the probability of selecting low-information redundant samples and successfully filters out noisy candidates. Furthermore, a comparison of the gray bounding boxes, which represent the hard-to-classify regions, reveals that IA-SFT strictly limits the selection of pseudo-labels within these ambiguous zones. IA-SFT maintains a relatively clear inter-class boundary and prevents the propagation of errors caused by a large number of mislabeled samples. These visualization results clearly demonstrate the effectiveness of IA-SFT in selecting high-value and reliable pseudo-labels.
> > >
> > > To further demonstrate the performance gain, we provide a comparison of the feature representations on the test set before and after applying the IA-SFT strategy, as shown in Fig. 2. This visualization highlights the improvement of the model's discriminative power.
> > >
> > > Comparing Fig. 2(a) and Fig. 2(b), there is a marked improvement in the separation of class clusters. Before IA-SFT, the features of changed (red) and unchanged (blue) samples are relatively dispersed, with noticeable overlap at the decision boundary. After IA-SFT fine-tuning, the two classes are mapped into more distinct and isolated manifolds. Additionally, it can be observed that the number of misclassified samples is also significantly reduced in regions far from the decision boundary. Thus, Fig. 2 demonstrates the effectiveness of IA-SFT-selected pseudo-labels in enhancing the model’s discriminative capability.
> > >
> > > These visualizations and analyses will be included in the appendix to further substantiate the effectiveness of our method and support our discussion. We hope that these additions will help address your concerns more thoroughly.

---

### Decision · Program_Chairs · 2026-04-30

**Decision:**

Accept (regular)

**Comment:**

This paper proposes a new semi-supervised fine-tuning framework for hyperspectral change detection that selects pseudo labels based on their influence on labeled data predictions rather than standard confidence scoring.

This paper receives three reviews, and the final scores are Weak accept, Weak accept, and Accept. The initial reviews raised several concerns, including clarity of originality & feature visualization (Reviewer UKyi), definition of “influence” & scope (Reviewer JYZc), and computational overhead & isolation of gains (TuNp). The reviewers were satisfied with the authors' rebuttal. However, there could be a minor  issue that the application domain is specific. While the method is architecture-agnostic, its immediate impact on the broader ICML community could be moderate.

Given that all the reviewers lean toward acceptance, I recommend Accept.